# KEYFRAMING THE FUTURE: DISCOVERING TEMPORAL HIERARCHY WITH KEYFRAME-INPAINTER PREDICTION

## ABSTRACT

To flexibly and efficiently reason about temporal sequences, abstract representations that compactly represent the important information in the sequence are needed. One way of constructing such representations is by focusing on the important events in a sequence. In this paper, we propose a model that learns both to discover such key events (or keyframes) as well as to represent the sequence in terms of them. We do so using a hierarchical Keyframe-Inpainter (KEYIN) model that first generates keyframes and their temporal placement and then inpaints the sequences between keyframes. We propose a fully differentiable formulation for efficiently learning the keyframe placement. We show that KEYIN finds informative keyframes in several datasets with diverse dynamics. When evaluated on a planning task, KEYIN outperforms other recent proposals for learning hierarchical representations.

## 1 INTRODUCTION

When thinking about the future, humans focus their thoughts on the important things that may happen (When will the plane depart?) without fretting about the minor details that fill each intervening moment (What is the last word I will say to the taxi driver?). Because the vast majority of elements in a temporal sequence contains redundant information, a temporal abstraction can make reasoning and planning both easier and more efficient. How can we build such an abstraction? Consider the example of a lead animator who wants to show what happens in the next scene of a cartoon. Before worrying about every low-level detail, the animator first sketches out the story by *keyframing*, drawing the moments in time when the important events occur. The scene can then be easily finished by other animators who fill in the rest of the sequence from the story laid out by the keyframes. In this paper, we argue that learning to discover such informative keyframes from raw sequences is an efficient and powerful way to learn to reason about the future.

Our goal is to learn such an abstraction for future image prediction. In contrast, much of the work on future image prediction has focused on frame-by-frame synthesis (Oh et al. (2015); Finn et al. (2016)). This strategy puts an equal emphasis on each frame, irrespective of the redundant content it may contain or its usefulness for reasoning relative to the other predicted frames. Other recent work has considered predictions that "jump" more than one step into the future, but these approaches either used fixed-offset jumps (Buesing et al., 2018) or used heuristics to select the predicted frames (Neitz et al., 2018; Jayaraman et al., 2019; Gregor et al., 2019). In this work, we propose a method that selects the keyframes that are most informative about the full sequence, so as to allow us to reason about the sequence holistically while only using a small subset of the frames. We do so by ensuring that the full sequence can be recovered from the keyframes with an *inpainting* strategy, similar to how a supporting animator finishes the story keyframed by the lead.

One possible application for a model that discovers informative keyframes is in long-horizon planning. Recently, predictive models have been employed for model-based planning and control (Ebert et al. (2018)). However, they reason about every single future time step, limiting their applicability to short horizon tasks. In contrast, we show that a model that reasons about the future using a small set of informative keyframes enables visual predictive planning for horizons much greater than previously possible by using keyframes as subgoals in a hierarchical planning framework.

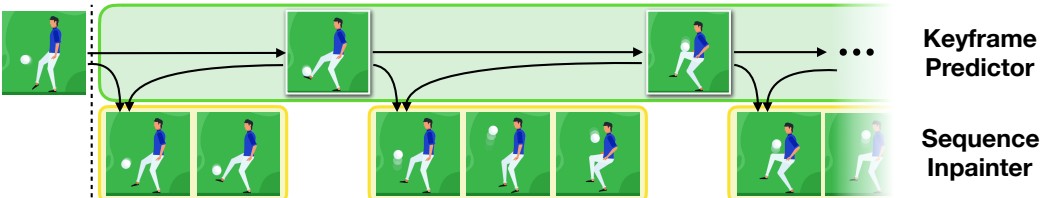

Figure 1: Keyframing the future. Instead of predicting one frame after the other, we propose to represent the sequence with the *keyframes* that depict the interesting moments of the sequence. The remaining frames can be inpainted given the keyframes.

To discover informative frames in raw sequence data, we formulate a hierarchical probabilistic model in which a sequence is represented by a subset of its frames (see Fig. 1). In this two-stage model, a keyframing module represents the keyframes as well as their temporal placement with stochastic latent variables. The images that occur at the timepoints between keyframes are then inferred by an inpainting module. We parametrize this model with a neural network and formulate a variational lower bound on the sequence log-likelihood. Optimizing the resulting objective leads to a model that discovers informative future keyframes that can be easily inpainted to predict the full future sequence.

Our contributions are as follows. We formulate a hierarchical approach for the discovery of informative keyframes using joint keyframing and inpainting (KEYIN). We propose a soft objective that allows us to train the model in a fully differentiable way. We first analyze our model on a simple dataset with stochastic dynamics in a controlled setting and show that it can reliably recover the underlying keyframe structure on visual data. We then show that our model discovers hierarchical temporal structure on more complex datasets of demonstrations: an egocentric gridworld environment and a simulated robotic pushing dataset, which is challenging for current approaches to visual planning. We demonstrate that the hierarchy discovered by KEYIN is useful for planning, and that the resulting approach outperforms other proposed hierarchical and non-hierarchical planning schemes on the pushing task. Specifically, we show that keyframes predicted by KEYIN can serve as useful subgoals that can be reached by a low-level planner, enabling long-horizon, hierarchical control.

## 2  RELATED WORK

**Hierarchical temporal structure.**    Hierarchical neural models for efficiently modeling sequences were proposed in Liu et al. (2015); Buesing et al. (2018). These approaches were further extended to predict with an adaptive step size so as to leverage the natural hierarchical structure in language data (Chung et al., 2016; Kádár et al., 2018). However, these models rely on autoregressive techniques for text generation and applying them to structured data, such as videos, might be impractical.

The video processing community has used keyframe representations as early as 1991 in the MPEG codec (Gall, 1991). Wu et al. (2018) adapted this algorithm in the context of neural compression; however, these approaches use constant offsets between keyframes and thus do not fully reflect the temporal structure of the data. Recently, several neural methods were proposed to leverage such temporal structure. Neitz et al. (2018) and Jayaraman et al. (2019) propose models that find and predict the least uncertain "bottleneck" frames. Gregor et al. (2019) construct a representation that can be used to predict any number of frames into the future. In contrast, we propose an approach for hierarchical video representation that discovers the keyframes that best describe a certain sequence.

In parallel to our work, Kipf et al. (2019) propose a related method for video segmentation via generative modeling. Kipf et al. (2019) focus on using the discovered task boundaries for training hierarchical RL agents, while we show that our model can be used to perform efficient hierarchical planning by representing the sequence with only a small set of keyframes. Also concurrently, Kim et al. (2019) propose a similar method to KEYIN for learning temporal abstractions. While Kim et al. (2019) focuses on learning hierarchical state-space models, we propose a model that operates directly in the observation space and performs joint keyframing and inpainting.

**Video modeling.** Early approaches to probabilistic video modeling include autoregressive models that factorize the distribution by considering pixels sequentially (Kalchbrenner et al., 2017; Reed et al., 2017). To reason about the images in the video holistically, latent variable approaches were developed based on variational inference (Chung et al., 2015; Rezende et al., 2014; Kingma & Welling, 2014), including (Babaeizadeh et al., 2018; Denton & Fergus, 2018; Lee et al., 2018) and large-scale models such as (Castrejon et al., 2019; Villegas et al., 2019). Kumar et al. (2019) is a recently proposed approach that uses exact inference based on normalizing flows (Dinh et al., 2014; Rezende & Mohamed, 2015). We build on existing video modeling approaches and show how they can be used to learn temporal abstractions with a novel keyframe-based generative model.

**Visual planning and model predictive control.** We build on recent work that explored applications of learned visual predictive models to planning and control. Several groups (Oh et al., 2015; Finn et al., 2016; Chiappa et al., 2017) have proposed models that predict the consequences of actions taken by an agent given its control output. Recent work (Byravan et al., 2017; Hafner et al., 2018; Ebert et al., 2018) has shown that visual model predictive control based on such models can be applied to a variety of different settings. In this work, we show that the hierarchical representation of a sequence in terms of keyframes improves planning performance in the hierarchical planning setting.

## 3    KEYFRAMING THE FUTURE

Our goal is to develop a model that generates sequences by first predicting key observations and the time steps when they occur and then filling in the remaining observations in between. To achieve this goal, in the following we (i) define a probabilistic model for joint keyframing and inpainting, and (ii) show how a maximum likelihood objective leads to the discovery of keyframe structure.

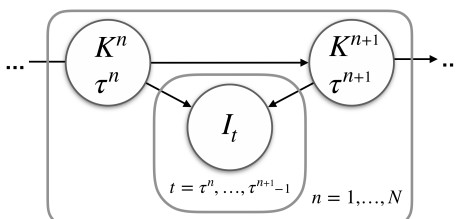

Figure 2: A probabilistic model for jointly keyframing and inpainting a future sequence. First, a sequence of keyframes $K^{1:N}$ is generated, as well as corresponding temporal indices $\tau^{1:N}$, defining the structure of the underlying sequence. In the second stage, for each pair of keyframes $K^n$ and $K^{n+1}$, the frames $I_{\tau^n:\tau^{n+1}-1}$ are inpainted.

### 3.1    A PROBABILISTIC MODEL FOR JOINT KEYFRAMING AND INPAINTING

We first describe a probabilistic model for joint keyframing and inpainting of a sequence $I_{1:T}$. The model consists of two parts: the *keyframe predictor* and the *sequence inpainter* (see Fig. 2).

The *keyframe predictor* takes in $C$ conditioning frames $I_{co}$ and produces $N$ keyframes $K^{1:N}$ as well as the corresponding time indices $\tau^{1:N}$:

$$p(K^{1:N}, \tau^{1:N}|I_{co}) = \prod_n p(K^n, \tau^n|K^{1:n-1}, \tau^{1:n-1}, I_{co}). \tag{1}$$

From each pair of keyframes, the *sequence inpainter* generates the sequence of frames in between:

$$p(I_{\tau^n:\tau^{n+1}-1}|K^n, K^{n+1}, \tau^{n+1} - \tau^n) = \prod_n p(I_t|K^n, K^{n+1}, I_{\tau^n:t-1}, \tau^{n+1} - \tau^n), \tag{2}$$

which completes the generation of the full sequence. The inpainter additionally observes the number of frames it needs to generate $\tau^{n+1} - \tau^n$. The temporal spacing of the most informative keyframes is data-dependent: shorter keyframe intervals might be required in cases of rapidly fluctuating motion, while longer intervals can be sufficient for steadier motion. Our model handles this by predicting the keyframe indices $\tau$ and inpainting $\tau^{n+1} - \tau^n$ frames between each pair of keyframes. We parametrize the prediction of $\tau^n$ in relative terms by predicting offsets $\delta^n$: $\tau^n = \tau^{n-1} + \delta^n$.

### 3.2    KEYFRAME DISCOVERY

To produce a complex multimodal distribution over $K$ we use a per-keyframe latent variable $z$ with prior distribution $p(z)$ and approximate posterior $q(z|I, I_{co})$.[1] We construct a variational lower bound

---

[1]For simplicity, the variable representing the full sequence is written without indices ($I$ is the same as $I^{1:T}$).

on the likelihood of both $I$ and $K$ as follows[2]:

$$\ln p(I, K|I_{co}) \geq \mathbb{E}_{q(z|I,I_{co})}\left[\sum_{n=1}^{N}\underbrace{\ln \mathbb{E}_{p(\tau^n,\tau^{n+1}|z^{1:n},I_{co})}\left[p(I_{\tau^n:\tau^{n+1}}|K^{n,n+1},\tau^{n+1}-\tau^n)\right]}_{\text{inpainting}}\right. \tag{3}$$
$$\left. + \underbrace{\ln p(K|z,I_{co})}_{\text{keyframing}}\right] - \underbrace{D_{\text{KL}}\left(q(z|I,I_{co})||p(z)\right)}_{\text{regularization}}.$$

In practice, we use a weight $\beta$ on the KL-divergence term, as is common in amortized variational inference (Higgins et al., 2017; Alemi et al., 2018; Denton & Fergus, 2018).

If a simple model is used for inpainting, most of the representational power of the model has to come from the keyframe predictor. We use a relatively powerful latent variable model for the keyframe predictor and a simpler Gaussian distribution produced with a neural network for inpainting. Because of this structure, the keyframe predictor has to predict keyframes that describe the underlying sequence well enough to allow a simpler inpainting process to maximize the likelihood. We will show that pairing a more flexible keyframe predictor with a simpler inpainter allows our model to discover semantically meaningful keyframes in video data.

## 4 CONTINUOUS RELAXATION BY LINEAR INTERPOLATION IN TIME

Our model can dynamically predict the keyframe placement $\tau^n$. However, learning a distribution over the discrete variable $\tau^n$ is challenging due to the expensive evaluation of the expectation over $p(\tau^n|z^{1:n}, I_{co})$ in the objective in Eq. 3. To be able to evaluate this term efficiently and in a differentiable manner while still learning the keyframe placement, we propose a continuous relaxation of the objective. The placement distribution $\tau^n$ defines a probability for each predicted frame to match to a certain frame in the ground truth sequence. Instead of sampling from this distribution to pick a target frame we produce a soft target for each predicted frame by computing the *expected* target frame, i.e. the weighted sum of all frames in the true sequence, each multiplied with the probability of matching to the predicted frame. When the entropy of $\tau^n$ converges to zero, the continuous relaxation objective is equivalent to the original, discrete objective. [3]

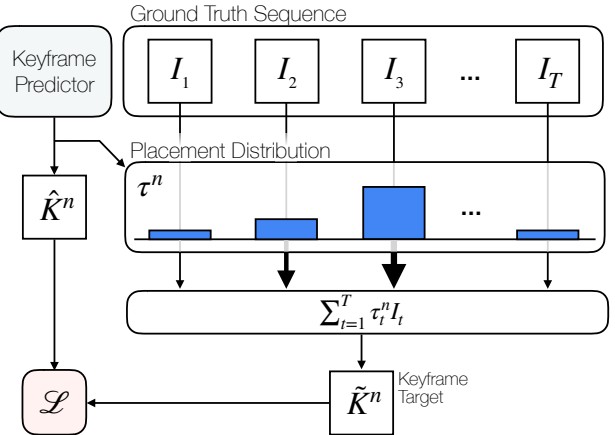

Figure 3: Soft keyframe loss in the relaxed formulation. For each predicted keyframe $\hat{K}^n$ we compute a target image $\tilde{K}^n$ as the sum of the ground truth images weighted with the corresponding distribution over index $\tau^n$. Finally, we compute the reconstruction loss between the estimated image $\hat{K}^n$ and the soft target $\tilde{K}^n$.

**Keyframe targets.** To produce a keyframe target, $\tilde{K}^n$, we linearly interpolate between the ground truth images according to the predicted distribution over the keyframe's temporal placement $\tau^n$: $\tilde{K}^n = \sum_t \tau_t^n I_t$, where $\tau_t^n$ is the probability that the $n^{th}$ keyframe occurs at timestep $t$. This process is depicted in Fig. 3.

We parametrize temporal placement prediction in terms of offsets $\delta$ with a maximum offset of $J$. Because of this, the maximum possible length of the predicted sequence is $NJ$. It is desirable for $J$ to be large enough to be able to capture the distribution of keyframes in the data, but this may lead to

---

[2]Keyframes K and time indices $\tau$ are modeled as independent *given* the corresponding latent variable $z$ such that the latter captures the dependency between the former two.

[3]We find this occurs most of the time in practice.

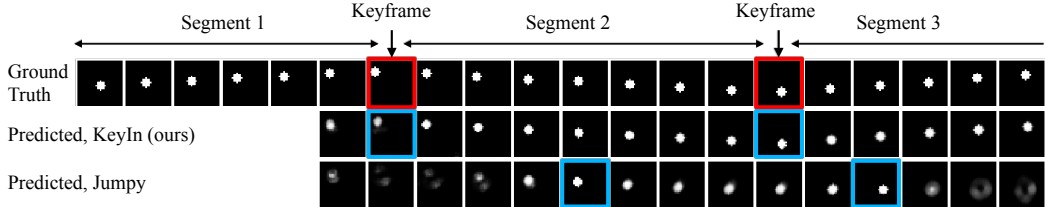

Figure 4: Sequences generated by KEYIN and a method with constant temporal keyframe offset (Jumpy) on Brownian Motion data. Generation is conditioned on the first five frames. The first half of the sequence is shown. Movement direction changes are marked red in the ground truth sequence and predicted keyframes are marked blue. We see that KEYIN can correctly reconstruct the motion as it selects an informative set of keyframes. The sequence generated by the Jumpy method does not reproduce the direction changes since they cannot be inferred from the selected keyframes.

the generation of sequences longer than the target $NJ > T$. To correctly compute the value of the relaxed objective in this case, we discard predicted frames at times $> T$ and normalize the placement probability output by the network so that it sums to one over the first $T$ steps. Specifically, for each keyframe we compute this probability as $c^n$: $c^n = \sum_{t \leq T} \tau_t^n$. The loss corresponding to the last two terms of Eq. (3) then becomes:

$$\mathcal{L}_{key} = \frac{\sum_n c^n \left( ||\hat{K}^n - \tilde{K}^n||^2 + \beta \, D_{\text{KL}} \left( q(z^n | I_{-C+1:T}, z^{1:n-1}) || p(z^n) \right) \right)}{\sum_n c^n}. \qquad (4)$$

**Inpainting targets.** To complete our relaxed objective, for each *ground truth* frame, we produce a target image composed from the inpainted frames.[4] We note that as offsets $\delta$ have a maximum range of $J$, and in general have non-zero probability on each timestep, the inpainting network needs to produce $J$ frames $\hat{I}_{1:J}^n$ between each pair of keyframes $\left( K^n, K^{n+1} \right)$. As in the previous section, the targets for ground truth images are given as an interpolation between generated images weighted by the probability of the predicted frame $\hat{I}_j^n$ being matched to ground truth frame $I_t$: $\tilde{I}_t = (\sum_{n,j} m_{j,t}^n \hat{I}_j^n) / \sum_{n,j} m_{j,t}^n$. Here, $m_{j,t}^n$ is the probability that the $j$-th predicted image in segment $n$ has an offset of $t$ from the beginning of the predicted sequence, which can be computed from $\tau^n$. To obtain a probability distribution over produced frames, we normalize the result with $\sum_{n,j} m_{j,t}^n$. The full loss for our model is:

$$\mathcal{L}_{total} = \mathcal{L}_{key} + \beta_I \sum_t ||I_t - \tilde{I}_t||^2. \qquad (5)$$

## 5 DEEP VIDEO KEYFRAMING

We show how to instantiate KEYIN with deep neural networks and train it on high-dimensional observations, such as images. We further describe an effective training procedure for KEYIN.

### 5.1 ARCHITECTURE

We use a common encoder-recurrent-decoder architecture (Denton & Fergus (2018); Hafner et al. (2018)). Video frames are first processed with a convolutional encoder module to produce image embeddings $\iota_t = \text{CNN}_{enc}(I_t)$. Inferred frame embeddings $\hat{\iota}$ are decoded with a convolutional decoder $\hat{I}_j^n = \text{CNN}_{dec}(\hat{\iota}_j^n)$. The keyframe predictor $p(K^{1:N}, \tau^{1:N} | z^{1:N}, I_{co})$ is parametrized with a Long Short-Term Memory network (LSTM, Hochreiter & Schmidhuber (1997)). To condition the keyframe predictor on past frames, we initialize its state with the final state of another LSTM that processes the conditioning frames. Similarly, we parametrize the sequence inpainter $p(I_{\tau^n:\tau^{n+1}} | K^n, K^{n+1}, \tau^{n+1} - \tau^n)$ with an LSTM. We condition the inpainting on both keyframe embeddings, $\hat{\kappa}^{n-1}$ and $\hat{\kappa}^n$, as well as the temporal offset between the two, $\delta^n$, by passing these inputs through a multi-layer perceptron that produces the initial state of the inpainting LSTM.

---

[4]This makes sure that each ground truth frame contributes equally to the final loss.

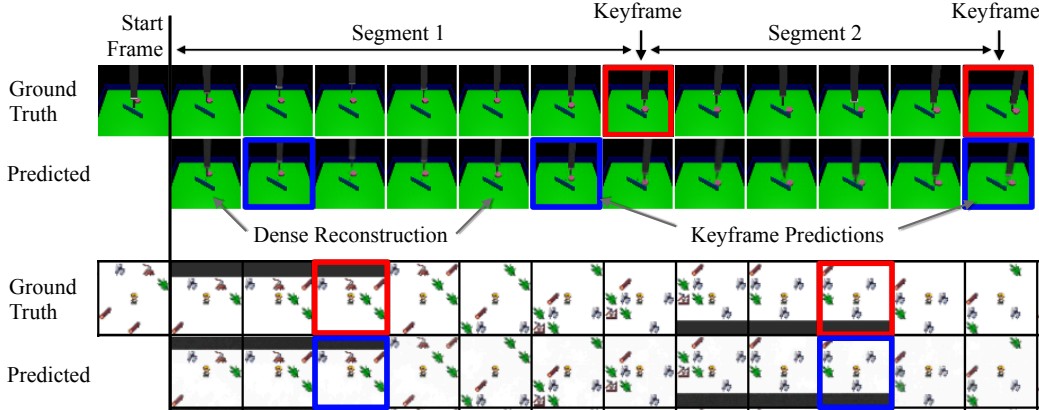

Figure 5: Example generations by KEYIN on (top) Pushing and (bottom) Gridworld data. The generation is conditioned on a single ground truth frame. Twelve of the 30 predicted frames are shown. We observe that for each transition between pushes and each action of the Gridworld agent our network predicts a keyframe either exactly at the timestep of the event or one timestep apart. Note, although agent position is randomized, objects not visible in the first image can be predicted in Gridworld because the maze is fixed across episodes.

We use a Gaussian distribution with identity variance as the output distribution for both the keyframe predictor and the inpainting model and a multinomial distribution for $\delta^n$. We parametrize the inference $q(z^{1:N}|I_{-C+1:T})$ with an LSTM with attention over the entire input sequence. The inference distribution is a diagonal covariance Gaussian, and the prior $p(z^{1:N})$ is a unit Gaussian. Further details of the inference procedure are given in Sec. B and Fig. 8 of the Appendix.

## 5.2 TRAINING PROCEDURE

We train our model in two stages. First, we train the sequence inpainter to inpaint between ground truth frames sampled with random offsets, thus learning interpolation strategies for a variety of different inputs. In the second stage, we train the keyframe predictor using the loss from Eq. 5 by feeding the predicted keyframe embeddings to the inpainter. In this stage, the weights of the inpainter are frozen and are only used to backpropagate errors to the rest of the model. We found that this simple two-stage procedure improves optimization of the model.

We use L1 reconstruction losses to train the keyframe predictor. We found that this and adding a reconstruction loss on the predicted embeddings of the keyframes, weighted with a factor $\beta_\kappa$, improved the ability of the model to produce informative keyframes. Target embeddings are computed using the same soft relaxation used for the target keyframes. More details of the loss computation are given in Sec. E and Algorithm 1 of the Appendix.

## 6 EXPERIMENTS

We evaluate the quality of KEYIN's representation for future sequences by addressing the following questions: (i) Can it discover and predict informative keyframes? (ii) Can it model complex data distributions? (iii) Is the discovered hierarchy useful for long-horizon hierarchical planning?

**Datasets.** We evaluate our model on three datasets containing structured long-term behavior. The *Structured Brownian motion* (SBM) dataset consists of binary image sequences of size $32 \times 32$ pixels in which a ball randomly changes directions after periods of straight movement of six to eight frames.

The *Gridworld Dataset* consists of 20k sequences of an agent traversing a maze with different objects. The agent sequentially navigates to objects and interacts with them following a task sketch. We use the same maze for all episodes and randomize the initial position of the agent and the task sketch. We use $64 \times 64$ pixel image observations and further increase visual complexity by constraining the field of view to a $5 \times 5$-cells egocentric window.

The *Pushing Dataset* consists of 50k sequences of a robot arm pushing a puck towards a goal on the opposite side of a wall. Each sequence consists of six consecutive pushes. We vary start and target position of the puck, as well as the placement of the wall. The demonstrations were generated with the MuJoCo simulator (Todorov et al., 2012) at a resolution of $64 \times 64$ pixels. For more details on the data generation process, see Sec.D of the Appendix.

Further details about the experimental setup are given in Sec. C of the Appendix.

## 6.1 KEYFRAME DISCOVERY

To evaluate KEYIN's ability to discover keyframes, we train KEYIN on all three datasets with $N = 6$, which can be interpreted as selecting the $N$ most informative frames from a sequence. We show qualitative examples of keyframe discovery for the SBM dataset in Fig. 4 and for the Gridworld and Pushing datasets in Fig. 5.

Table 1: F1 accuracy score for keyframe discovery on all three datasets. Higher is better.

| METHOD | BROWNIAN | PUSH | GRIDWORLD |
|---|---|---|---|
| RANDOM | 0.15 | 0.18 | 0.12 |
| STATIC | 0.21 | 0.18 | 0.25 |
| SURPRISE | 0.73 | 0.10 | 0.32 |
| KEYIN (OURS) | **0.94** | **0.43** | **0.42** |

On all datasets the model discovers meaningful keyframes which mark direction changes of the ball, transitions between pushes or interactions with objects, adapting its keyframe prediction patterns to the data. Consequently, the inpainter network is able to produce frames of high visual quality. Misplaced keyframes yield blurry interpolations, as can be seen for the jumpy prediction in Fig. 4. This suggests that keyframes found by KEYIN describe the overall sequences better.

To show that KEYIN discovers informative keyframes, we compare keyframe predictions against an alternative approach that measures the surprise associated with observing a frame given the previous frames. This approach selects keyframes as the $N$ frames with the largest peaks in "surprise" as measured by the KL-divergence $D_{\mathrm{KL}}[q(z_t|I_{1:t})||p(z_t)]$ between the prior and the posterior of a stochastic predictor based on Denton & Fergus (2018) (see Sec. F and Algorithm 2 of the Appendix for details). We provide comparisons to alternative formulations of surprise in Appendix Sec. F, Tab. 3.

For quantitative analysis, we define approximate ground truth keyframes to be the points of direction change for the SBM dataset, the moments when the robot lifts its arm to transitions between pushes, or when the agent interacts with objects in the gridworld. We report F1 scores that capture both the precision and recall of keyframe discovery.

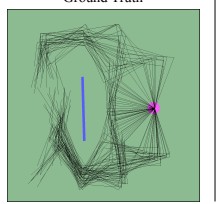 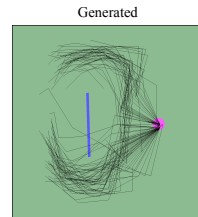

Ground Truth          Generated

Figure 6: Distribution of trajectories sampled from KEYIN. Each black line denotes one of 100 trajectories of the manipulated object. The obstacle is shown in blue and the initial position in pink. We see that our model covers both modes of the distribution, producing both trajectories that go to the right and to the left of the obstacle.

We additionally compare to random keyframe placement, and a learned but static baseline that is the same for all sequences. The evaluation in Tab. 1 shows that KEYIN discovers better keyframes than alternative methods. The difference is especially large on the more complex Pushing and Gridworld datasets. The surprise-based method does not reason about which frames are most helpful to reconstruct the entire trajectory and thus is unable to discover the correct structure on the more complex datasets. In addition to the F1 scores, we report temporal distance between predicted and annotated keyframes in Appendix, Tab. 4, also indicating that KEYIN is better able to discover the temporal structure in both datasets.

## 6.2 KEYFRAME-BASED VIDEO MODELING

Even though the focus of this work is on discovering temporal structure via keyframing and not on improving video prediction quality, we verify that KEYIN can represent complex data distributions in terms of discovered keyframes and attains high diversity and visual quality. We show sample generations from our model on the Pushing and Gridworld datasets on the supplementary website[5].

---

[5] https://sites.google.com/view/keyin

We see that KeyIn is able to faithfully model complex distributions of video sequences. We further visualize multiple sampled Pushing sequences from our model conditioned on the same start position in Fig. 6, showing that KEYIN is able to cover both modes of the demonstration distribution. We further show that KEYIN compares favorably to prior work on video prediction metrics on sequence modeling in Tab. 5 of the Appendix, and outperforms prior approaches in terms of keyframe modeling in Appendix, Tab. 6.

## 6.3 ROBUSTNESS OF KEYFRAME DETECTION

In the previous sections, we showed that when the sequence can indeed be summarized with $N$ keyframes, KEYIN predicts the keyframes that correspond to our notion of salient frames. However, what happens if we train KEYIN to select a larger or a smaller amount of keyframes?

To evaluate this, we measure KEYIN recall with extra and precision with fewer available keyframes. We note that high precision is unachievable in the first case and high recall is unachievable in the second case, since these problems are misspecified. As these numbers are not

Table 2: Keyframe discovery for varied number of predicted keyframes. The data has approximately 6 keyframes. Uninterpretable entries are omitted for clarity: see the text for details.

| # KEYFRAMES | | 4 | 6 | 8 |
|---|---|---|---|---|
| BROWNIAN | PRECISION | 0.92 | 0.92 | - |
| | RECALL | - | 0.96 | 0.90 |
| PUSH | PRECISION | 0.30 | 0.38 | - |
| | RECALL | - | 0.48 | 0.46 |
| GRIDWORLD | PRECISION | 0.37 | 0.43 | - |
| | RECALL | - | 0.41 | 0.40 |

informative, we do not report them. In Tab. 2, we see that KEYIN is able to find informative keyframes even when $N$ does not exactly match the structure of the data. We further qualitatively show that KEYIN selects a superset or a subset of the original keyframes respectively in Sec. G. This underlines that our method's ability to discover keyframe structure is robust to the choice of the number of predicted keyframes.

As a first step towards analyzing the robustness of KEYIN under more realistic conditions we report keyframe discovery when trained and tested on sequences with additive Gaussian noise, a noise characteristic commonly found in real-world camera sensors. We find that KEYIN is still able to discover the temporal structure on both the Pushing and the Gridworld dataset. For qualitative and quantitative results, see Appendix Fig. 11 and Tab. 7.

## 6.4 HIERARCHICAL KEYFRAME-BASED PLANNING

We have seen that KEYIN can find frames that correspond to an intuitive notion of keyframes. This demonstrates that the keyframes discovered by KEYIN do indeed capture an abstraction that compactly describes the sequence. In light of this, we hypothesize that an informative set of keyframes contains sufficient information about a sequence to effectively follow the trajectory it shows. To test this, we use the inferred keyframes as subgoals for hierarchical planning in the pushing environment. During task execution, we first plan a sequence of keyframes that reaches the target using our learned keyframe predictor. Specifically, we generate keyframe trajectories from our model by sampling latent variables $z$ from the prior and using them to roll out the keyframe prediction model. We optimize for a sequence of latent variables $z$ that results in a keyframe trajectory which reaches the goal using the Cross-Entropy Method (CEM, Rubinstein & Kroese (2004)). We then execute the plan by using the keyframes as subgoals for a low-level planner. This planner reaches each subgoal via model predictive control using ground truth dynamics, again employing CEM for optimization of the action trajectory. This planning procedure is illustrated in Fig. 7 (left). For more details, see Sec. I and Algs. 3 and 4 of the Appendix.

We find that KEYIN is able to plan coherent subgoal paths towards the final goal that often lead to successful task execution (executions are shown on the supplementary website[6]). To quantitatively evaluate the keyframes discovered, we compare to alternative subgoal selection schemes: fixed time offset (*Jumpy*, similar to Buesing et al. (2018)), a method that determines points of peak surprise (*Surprise*, see Sec. 6.1), and a bottleneck-based subgoal predictor (time-agnostic prediction or TAP, Jayaraman et al. (2019)). We additionally compare to an approach that plans directly towards the final goal using the low-level planner (*Flat*). We evaluate all methods with the shortest path between

---

[6]https://sites.google.com/view/keyin

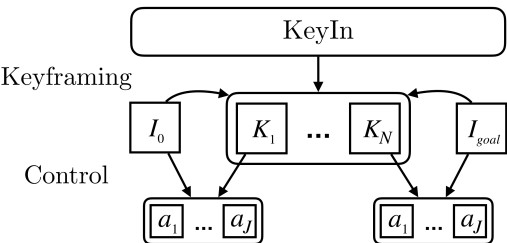

Keyframing

Control

| METHOD | POSITION ERROR | SUCCESS RATE |
|---|---|---|
| INTITIAL | $1.32 \pm 0.06$ | - |
| RANDOM | $1.32 \pm 0.07$ | - |
| FLAT | $0.90 \pm 0.14$ | 15.0 % |
| TAP | $0.80 \pm 0.16$ | 23.3 % |
| SURPRISE | $0.64 \pm 0.28$ | 50.8 % |
| JUMPY | $0.62 \pm 0.33$ | 58.8 % |
| KEYIN (OURS) | $\mathbf{0.50 \pm 0.26}$ | **64.2 %** |

Figure 7: Hierarchical planning on the Pushing dataset. **Left**: We use the model to produce keyframes that represent the sequence between the current observation image and the goal. A low-level planner based on model predictive control produces the actions, $a_t$, executed to reach each keyframe, until the final goal is reached. **Right**: Planning performance on a Pushing task. The hierarchy discovered by KEYIN outperforms comparable planning approaches.

the target and the actual position of the object after the plan is executed. All compared methods use the same low-level planner as we only want to measure the quality of the predicted subgoals.

As shown in Fig. 7 (right), our method outperforms all prior approaches. TAP shows only a moderate increase in performance over the Flat planner, which we attribute to the fact that it fails to predict good subgoals and often simply predicts the final image as the bottleneck. This is likely due to the relatively large stochasticity of our dataset and the absence of the clear bottlenecks that TAP is designed to find. Our method outperforms the planners that use Jumpy and Surprise subgoals. This further confirms that KEYIN is able to produce keyframes that are informative about the underlying trajectory, such that planning toward these keyframes makes it easier to follow the trajectory.

## 7 DISCUSSION

We presented KEYIN, a method for representing a sequence by its informative keyframes by jointly keyframing and inpainting. KEYIN first generates the keyframes of a sequence and their temporal placement and then produces the full sequence by inpainting between keyframes. We showed that KEYIN discovers informative keyframes on several datasets with stochastic dynamics. Furthermore, by using the keyframes for planning, we showed our method outperforms several other hierarchical planning schemes. Our method opens several avenues for future work. First, an improved training procedure that allows end-to-end training is desirable. Second, more powerful hierarchical planning approaches can be designed using the keyframe representation to scale to long-term real-world tasks. Finally, the proposed keyframing method can be applied to a variety of applications, including video summarization, video understanding, and multi-stage hierarchical video prediction.

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

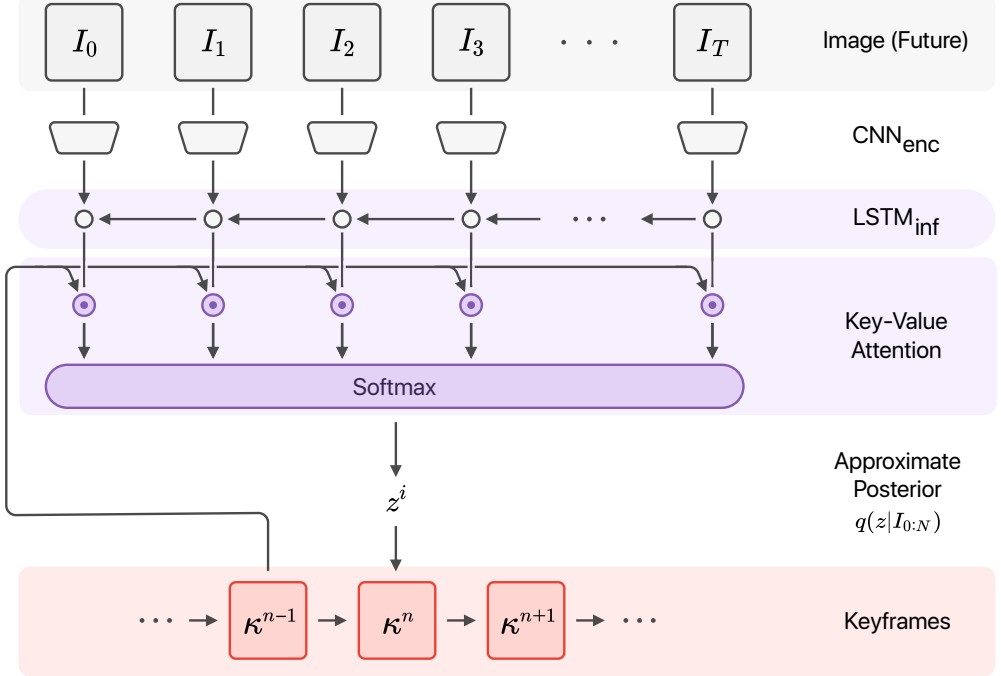

Figure 8: Structure of the keyframe inference network. This diagram depicts the procedure to infer the embedding of the $n$-th keyframe, $\kappa^n$, given the previously inferred keyframe embedding $\kappa^{n-1}$ and the future images. The initial state of $\text{LSTM}_{\text{key}}$ is produced by $\text{LSTM}_{\text{cond}}$ (not shown), which takes the embedding of past images as input. This ensures that the See the text for more details.

## A   VIDEOS

We include video results on the supplementary website at `https://sites.google.com/view/keyin`. The website includes inference samples, prior samples, and exectued trajectories for all methods.

## B   ARCHITECTURE DETAILS

We found simple attention over LSTM outputs to be an effective inference procedure. Our approximate inference network $\text{LSTM}_{inf}$ outputs $(\kappa_t^{inf}, \zeta_t)_{t \leq T}$, where $\kappa^{inf}$ is an embedding used to compute an attention weight and the $\zeta_t$ are values to be attended over. We compute the posterior distribution over $z^t$ using a key-value attention mechanism (Bahdanau et al., 2015; Luong et al., 2015):

$$a_{n,t} = \exp(d(\hat{\kappa}^{n-1}, \kappa_t^{inf})) \tag{6}$$

$$\mu^n, \sigma^n = (\sum_t a_{n,t}\zeta_t)/\sum_t a_{n,t}. \tag{7}$$

The distance metric, $d$, is the standard inner product. The architecture used for keyframe inference, including the attention mechanism, is depicted in Supplemental Fig. 8.

## C   EXPERIMENTAL SETUP

We set the prediction horizon to $T = 30$ frames and predict $N = 6$ segments with $J = 10$ frames each for the SBM dataset and 6 frames each for the Pushing dataset. We pre-train the interpolator on segments of two to eight frames for Structured Brownian motion data, and two to six frames for Pushing data. The weight on the KL-divergence term for the interpolator VAE is $1e-3$. For training

the keyframe predictor, we set $\beta_K = 0, \beta_\kappa = 1, \beta = 5\text{e}{-}2$. The hyperparameters were hand-tuned. We activate the generated images with a sigmoid and use BCE losses on each color channel to avoid saturation. The convolutional encoder and decoder both have three layers for the Structured Brownian motion dataset and four layers for the Pushing dataset. We use a simple two-layer LSTM with a 256-dimensional state in each layer for all recurrent modules. Each LSTM has a linear projection layer before and after it that projects the observations to and from the correct dimensions. We use the Adam optimizer (Kingma & Ba, 2015) with $\beta_1 = 0.9$ and $\beta_2 = 0.999$, batch size of 30, and a learning rate of $2\text{e}{-}4$. For more details please refer to the appendix. Each network was trained on a single high-end NVIDIA GPU. We trained the interpolator for 100K iterations, and the keyframe predictor for 200K iterations. The toal training time took about a day.

In the Pushing environment, we use a held-out test set of 120 of sequences. The Structured Brownian Motion dataset is generated automatically and is potentially infinite. We used 1000 testing samples on the Structured Brownian Motion generated using a different random seed.

## D  DATA COLLECTION IN THE MUJOCO ENVIRONMENT

The data collection for our pushing dataset was performed in an environment simulated in MuJoCo Todorov et al. (2012). In the environment, a robot arm initialized at the center of the table pushes an object to a goal position located at the other side of a wall-shaped obstacle.

The demonstrations followed a rule-based algorithm that first samples subgoals between the initial position of the object and the goal and then runs a deterministic pushing procedure to the subgoals in order. The ground truth keyframes of the demonstrations were defined by frames at which subgoals were completed.

We subsampled demonstration videos by a factor of two when saving them to the dataset, dropping every other frame in the trajectory and averaging actions of every two consecutive frames. For all datasets we generated for this environment following a rule-based algorithm, we only kept successful demonstrations and dropped the ones that fail to push the object to the goal position within a predefined horizon.

## E  DETAILS OF THE LOSS COMPUTATION ALGORITHM

We describe the details of the continuous relaxation loss computation in Algorithm 1.

Note that we efficiently implement computing of the cumulative distributions $\tau$ as a convolution, which allows us to vectorize much of the computation. Computational complexity of the proposed implementation scales linearly with the number of keyframes $N$, and number of allowed frames per segment $J$ and number of ground truth frames $T$. The final complexity is $\mathcal{O}(NTJ)$, which we find in practice to be negligible compared to the time needed for the forward and backward pass.

## F  SURPRISE BASELINE

Standard stochastic video prediction methods do not attempt to estimate keyframes, as they are designed to densely estimate future videos frame-by-frame. Accordingly, they cannot be used directly as baselines for keyframe prediction methods, such as KEYIN. However, Denton & Fergus (2018) observe that the variance of the learned prior of a stochastic video prediction model tends to spike before an uncertain event happens. We exploit this observation to find the points of high uncertainty for our strong Surprise baseline. We use the KL divergence between the prior and the approximate posterior $\text{KL}[q(z_t|I_{1:t})||p(z_t)]$ to measure the surprise. This quantity can be interpreted as the number of bits needed to encode the latent variable describing the next state, it will be larger if the next state is more stochastic.

We train a stochastic video prediction network with a fixed prior (SVG-FP, Denton & Fergus (2018)) with the same architectures of encoder, decoder, and LSTM as our model. We found that selecting the peaks of suprise works the best for finding true keyframes. The procedure we use to select the keyframes is described in Algorithm 2. In order to find the keyframes in a sequence sampled from the prior, we run the inference network on the generated sequence.

---

**Algorithm 1** Continuous relaxation loss computation

**Parameters:** Number of ground truth frames $T$, Number of keyframes $N$

**Input:** Ground truth frames $I_{1:T}$, Generated frames $\hat{I}_i^t$, generated offset distributions $\delta^n$

Convert the distributions of interframe offsets $\delta^n$ to keyframe timesteps $\tau^n$. For the first keyframe, $\tau^1 = \delta^1$.

**for** $t = 2 \dots M$ **do**

  Compute further $\tau^n$ with chain rule. This can be efficiently computed via convolution:

$$\tau^n = \tau^{n-1} * \delta^n, \text{ i.e. } \tau_t^n = \sum_j \tau_{n-j+1}^{n-1} \delta_j^n.$$

**end for**

Compute probabilities of keyframes being within the predicted sequence: $c^n = \sum_{t \leq T} \tau_t^n$.

Compute soft keyframe targets: $\tilde{K}^n = \sum_t \tau_t^n I_t$.

Compute the keyframe loss: $\left( \sum_n c^n ||\hat{K}^n - \tilde{K}^n||^2 \right) / \sum_n c^n$.

Get probabilities of segments ending after particular frames: $e_j^n = \sum_{j>i} \delta_j^n$.

Get distributions of individual frames timesteps: $m_{j,t}^n \propto \tau_{t-j+1}^{n-1} e_j^n$.

Compute soft individual frames: $\tilde{I}_t = \sum_{t,i} m_{j,t}^n \hat{I}_i^t$

Compute the sequence loss: $\sum_t ||I_t - \tilde{I}_t||^2$.

---

**Algorithm 2** Selecting keyframes via Surprise

**Parameters:** Number of ground truth frames $N$, Desired number of keyframes $M$

**Input:** Input sequence $I_{1:T}$, Stochastic Video Prediction model $SVG(.)$

Run the inference network over the sequence: $q(z_{1:T}|I_{1:T}) = SVG(I_{1:T})$.

Get the surprise measure: $s_t = KL[q(z_t|I_{1:t})||p(z_t)]$.

Find the set of peak surprise points $S$ where: $s_t > s_{t+1} \wedge s_t < s_{t-1}$.

**if** $|S| < M$ **then**

  add $M - |S|$ maximum surprise points to S.

**end if**

**Return:** The $M$ keyframes from $S$ with maximum surprise.

---

## G    ABLATION OF THE NUMBER OF PREDICTED KEYFRAMES

We show qualitative results of training KEYIN with $N = 4, 6$ (optimal number), and 8 on the SBM dataset in Fig. 9. We observe that if we train KEYIN to select a smaller or a larger number of keyframes than needed, it learns to predict a subset or a superset of the true keyframes, respectively. This property follows from the structure of the model, which encourages the model to predict the keyframes that allow the full sequence to be inpainted. When too few keyframes are available, the model will be unable to put keyframes at all important times, but those it picks must still be good for inpainting. When more keyframes are available than necessary, the model can place the additional keyframes at any time, as only a subset of the keyframes are needed to ensure good inpainting.

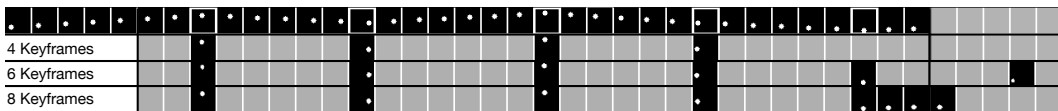

Figure 9: Qualitative keyframe discovery on the Structured Brownian Motion dataset for varying number of predicted keyframes. **Top**: Ground truth sequence, keyframes with bold white frame. **Bottom**: KEYIN keyframe predictions at their predicted temporal placement. Even if the number of predicted keyframes does not match the true number of keyframes KEYIN correctly discovers the keyframes and their temporal placement.

Table 3: We compare different formulations for a surprise-based keyframe detection method: (1) detecting maxima of the KL divergence between prior and posterior in a stochastic prediction model, (2) detecting maxima in the lower bound on data likelihood $\log p$ (ELBO) of a stochastic prediction model, (3) the formulation proposed in Denton & Fergus (2018) that detects maxima of the variance of a learned prior distribution.

| DATASET | PUSH | | | GRIDWORLD | | |
|---|---|---|---|---|---|---|
| METHOD | F1 ↑ | min $d_{\text{KF}}^{\text{TRUE}}$ ↓ | min $d_{\text{KF}}^{\text{PRED}}$ ↓ | F1 ↑ | min $d_{\text{KF}}^{\text{TRUE}}$ ↓ | min $d_{\text{KF}}^{\text{PRED}}$ ↓ |
| KL-SURPRISE | **0.25** | **1.44** | **2.05** | 0.32 | 1.42 | 1.86 |
| $\log p$-SURPRISE | **0.24** | **1.42** | **2.08** | 0.31 | 1.45 | 1.83 |
| DENTON & FERGUS (2018) | 0.17 | 1.73 | 2.01 | **0.35** | **1.10** | **1.53** |

Table 4: In addition to the F1 scores we report the minimal temporal distance to the next keyframe as an additional metric that is more graceful with respect to "close misses". Specifically, we report the distance to the next annotated keyframe averaged across predicted keyframes, min $d_{\text{KF}}^{\text{true}}$, and, inversely, the distance to the next predicted keyframe for each annotated keyframe, min $d_{\text{KF}}^{\text{pred}}$. For both datasets the distance metrics support the F1 results: KEYIN discovers keyframes that are better aligned with the annotated keyframes than the baselines.

| DATASET | PUSH | | | GRIDWORLD | | |
|---|---|---|---|---|---|---|
| METHOD | F1 ↑ | min $d_{\text{KF}}^{\text{TRUE}}$ ↓ | min $d_{\text{KF}}^{\text{PRED}}$ ↓ | F1 ↑ | min $d_{\text{KF}}^{\text{TRUE}}$ ↓ | min $d_{\text{KF}}^{\text{PRED}}$ ↓ |
| STATIC | 0.18 | 1.67 | **1.25** | 0.25 | 1.22 | 1.07 |
| SURPRISE | 0.25 | 1.44 | 2.05 | 0.32 | 1.42 | 1.86 |
| KEYIN (OURS) | **0.43** | **1.25** | 1.86 | **0.42** | **1.03** | **0.99** |

## H  VIDEO MODELING PERFORMANCE

We further report quantitative results on standard video prediction metrics, Structural Similarity Index (SSIM) and Peak Signal-to-Noise Ratio (PSNR), in Tab. 5. KEYIN is able to match performance of two comparable prior approaches, showing that the keyframe-based modeling is able to represent complex data distributions.

## I  PLANNING ALGORITHM

---
**Algorithm 3** Planning in the subgoal space.

---
**Input:** Keyframe model $\text{KEYIN}(.,.)$, cost function $c$
**Input:** Start and target images $I_1$ and $I_{\text{target}}$
Set the sampling distribution to the prior:
      $\mu_i = 0, \sigma_i = I$
**for** $i = 1 \dots H$ **do**
    Sample $L$ sequences of latent variables:
        $z^{1:N} \sim \mathcal{N}(\mu_i, \sigma_i)$
    Produce $L$ subgoal plans: $\hat{K}^{1:N} = \text{KEYIN}(I_1, z^{1:N})$
    Compute cost between generated and true targets:
        $c(\hat{K}^N, I_{\text{target}})$
    Choose $L'$ best plans, refit sampling distribution:
        $\mu_{i+1}, \sigma_{i+1} = \text{fit}(z_i')$
**end for**
**Return:** Best subgoal plan $K^{1:N}$

---

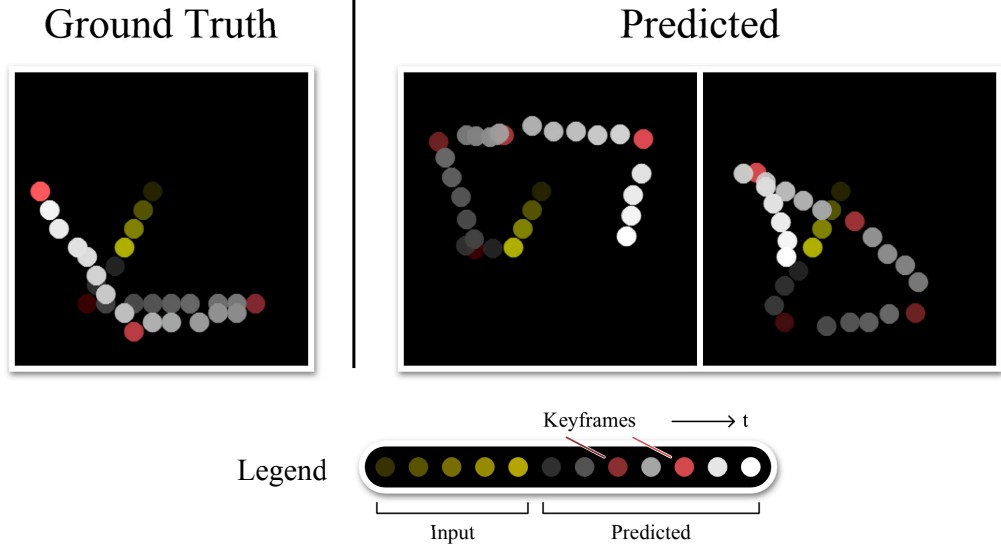

Figure 10: A training sequence and two samples from our model on the Structured Brownian motion dataset. Each image shows an entire trajectory. Our model first samples the keyframes (shown in red), and then deterministically predicts the rest of the sequence. The image resolution was enhanced for viewability.

Table 5: SSIM and PSNR scores on pushing and gridworld dataset. Higher is better.

| DATASET | PUSH | | GRIDWORLD | |
|---|---|---|---|---|
| METHOD | PSNR | SSIM | PSNR | SSIM |
| DENTON & FERGUS (2018) | $33.3 \pm 0.1$ | $0.956 \pm 0.001$ | $29.4 \pm 0.1$ | $0.812 \pm 0.01$ |
| JUMPY | $33.7 \pm 0.1$ | $0.960 \pm 0.001$ | $29.9 \pm 0.1$ | $0.831 \pm 0.001$ |
| KEYIN (OURS) | $33.4 \pm 0.7$ | $0.959 \pm 0.001$ | $29.3 \pm 0.1$ | $0.820 \pm 0.001$ |

To apply the KEYIN model for planning, we use the approach for visual planning outlined in Algorithm 1. At the initial timestep, we use the cross-entropy method (CEM) Rubinstein & Kroese (2004) to select subgoals for the task. To do so, we sample $\tilde{M}$ latent sequences $z_0$ from

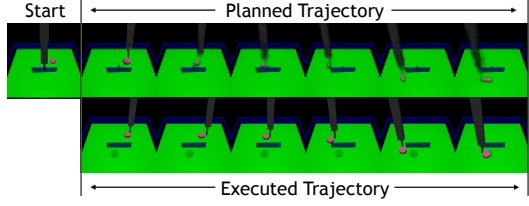

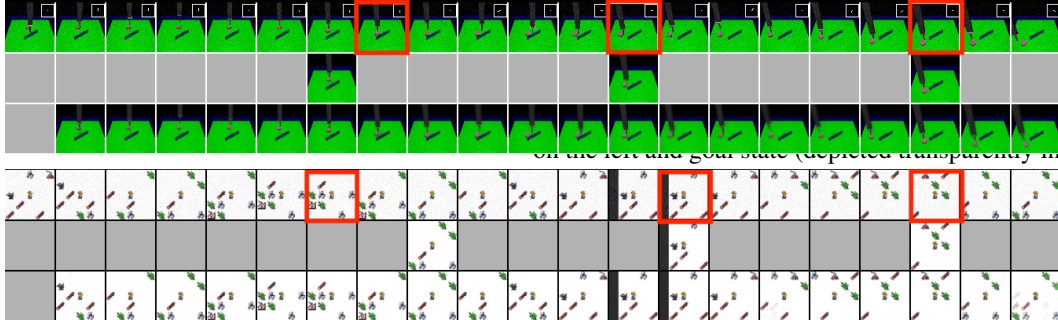

Figure 11: Example keyframe detections on noisy sequences. Red frames mark annotated keyframes. **Top**: Pushing dataset. **Bottom**: Gridworld dataset. Each triplet depicts *top*: ground truth sequence with additive Gaussian noise, *middle*: predicted keyframes at the predicted time steps, *bottom*: predicted full sequence. KEYIN is reliably able to detect keyframes and reconstruct the full sequence.

Table 6: **Keyframe** SSIM and PSNR scores on pushing and gridworld dataset. Higher is better.

| Dataset | Push | | Gridworld | |
|---|---|---|---|---|
| Method | PSNR | SSIM | PSNR | SSIM |
| Jumpy | $28.25 \pm 0.11$ | $0.904 \pm 0.001$ | $16.3 \pm 0.17$ | $\mathbf{0.632 \pm 0.001}$ |
| KeyIn (Ours) | $\mathbf{29.5 \pm 0.16}$ | $\mathbf{0.911 \pm 0.001}$ | $\mathbf{18.4 \pm 0.17}$ | $\mathbf{0.636 \pm 0.001}$ |

Table 7: F1 score and distance to closest annotated / predicted keyframe when trained and tested on sequences with additive Gaussian noise. KeyIn is able to reliably find keyframes on both datasets even when trained and tested on noisy sequences. Even though the F1 score is lower on the Pushing dataset, the distances indicate that the discovered keyframes are well aligned with the annotated keyframes even under noise.

| Dataset | Push | | | Gridworld | | |
|---|---|---|---|---|---|---|
| Method | F1 $\uparrow$ | min $d_{\mathrm{KF}}^{\mathrm{TRUE}}$ $\downarrow$ | min $d_{\mathrm{KF}}^{\mathrm{PRED}}$ $\downarrow$ | F1 $\uparrow$ | min $d_{\mathrm{KF}}^{\mathrm{TRUE}}$ $\downarrow$ | min $d_{\mathrm{KF}}^{\mathrm{PRED}}$ $\downarrow$ |
| KeyIn, no-noise | 0.43 | 1.25 | 1.86 | 0.42 | 1.03 | 0.99 |
| KeyIn, Gauss-noise | 0.25 | 1.21 | 1.34 | 0.43 | 1.00 | 0.96 |

the prior $\mathcal{N}(0, I)$ and use the keyframe model to retrieve $\tilde{M}$ corresponding keyframe sequences, each with $L$ frames. We define the cost of an image trajectory as the distance between the target image and the final image of each keyframe sequence defined under a domain-specific distance function (see below). In the update step of the CEM algorithm, we rank the trajectories based on their cost and fit a diagonal Gaussian distribution to the latents $\boldsymbol{z}'$ that generated the $\tilde{M}' = r\tilde{M}$ best sequences, where $r$ is the elite ratio. We repeat the procedure above for a total of $N$ iterations.

We define the cost between two frames used during planning as the Euclidean distance between the center pixels of the object in both frames. We recover the center pixel via color-based segmentation of the object. While this cost function is designed for the particular planning environment we are testing on, our algorithm can be easily extended to use alternative, more domain-agnostic cost formulations that have been proposed in the literature Finn & Levine (2017); Ebert et al. (2017; 2018).

After subgoals are selected, we use a CEM based planner to produce rollout trajectories. Similar to the subgoal generation procedure, at each time step, we initially sample $M$ action sequences $\boldsymbol{u}_0$ from the prior $\mathcal{N}(0, I)$ and use the ground truth dynamics of the simulator to retrieve $M$ corresponding image sequences, each with $l$ frames[7]. We define the cost of an image trajectory as the distance between the target image and the final image of each trajectory. In the update step, we rank the trajectories based on their cost and fit a diagonal Gaussian distribution to the actions $\boldsymbol{u}'$ that generated the $M' = rM$ best sequences. After sampling a new set of actions $\boldsymbol{u}_{n+1}$ from the fitted Gaussian distributions we repeat the procedure above for a total of $N$ iterations.

Finally, we execute the first action in the action sequence corresponding to the best rollout of the final CEM iteration. The action at the next time step is chosen using the same procedure with the next observation as input and reinitialized action distributions. The algorithm terminates when the specified maximal number of planning steps $T_{\mathrm{max}}$ has been executed or the distance to the goal is below a set threshold.

---

[7]In practice, we clip the sampled actions to a maximal action range $[-a_{\max}, +a_{\max}]$ before passing them to the simulator.

Table 8: Hyperparameters for the visual planning experiments.

| planning Parameters | |
|---|---|
| Max. planning timesteps ($T_{\max}$) | 60 |
| Max. per subgoal timesteps ($T_{s,\max}$) | 10 |
| Keyframe prediction horizon ($L$) | 6 |
| # keyframe sequences ($\tilde{M}$) | 200 |
| planning horizon ($l$) | 8 |
| # planning sequences ($M$) | 200 |
| Elite fraction ($r = M'/M$) | 0.05 |
| # refit iterations ($N$) | 3 |
| max. action ($a_{\max}$) | 1.0 |
| $d_{\text{switch}}$ | 5 |

We switch between planned subgoals if (i) the subgoal is reached, i.e. the distance to the subgoal is below a threshold $d_{\text{switch}}$ measured in pixels, or (ii) the current subgoal was not reached for $T_{s,\max}$ execution steps. We use the true goal image as an additional, final subgoal.

---

**Algorithm 4** Full two-stage planning algorithm

---

**Input:** Keyframe model $\hat{K}_{1:L} = \text{LSTM}_{key}(I, z_{1:L})$
**Input:** Video prediction model $\hat{I}_{t:t+l} = \text{LSTM}_{inter}(I_{1:t-1}, u_{2:t+l})$
**Input:** Subgoal index update heuristics $\text{ix}_{t+1} = f(\text{ix}_t, I_t, K_{1:L})$
**Input:** Start and goal images $I_1$ and $I_{\text{goal}}$.
Initialize latents from prior: $\boldsymbol{z}_0 \sim \mathcal{N}(0, I)$.
**for** $i = 1 \dots N_{it}$ **do**
    Rollout keyframe model for $L$ steps, obtain $\tilde{M}$ future keyframe sequences $\hat{\boldsymbol{K}}_{1:L}$.
    Compute distance between final and goal image: $c = \text{dist}(\hat{K}_L, I_{\text{goal}})$.
    Choose $\tilde{M}'$ best sequences, refit Gaussian distribution: $\boldsymbol{\mu}_{i+1}^{key}, \boldsymbol{\sigma}_{i+1}^{key} = \text{fit}(\boldsymbol{K}_i')$.
    Sample new latents from updated distribution: $\boldsymbol{z}_{i+1} \sim \mathcal{N}(\boldsymbol{\mu}_{i+1}^{key}, \boldsymbol{\sigma}_{i+1}^{key})$.
**end for**
Feed best sequence of latents into keyframe model to obtain subgoals: $K_{1:L}^* = \text{LSTM}_{key}(I_1, z_{N_{it},0}^*)$.
Set current subgoal to $\text{ix}_1 = 1$.
**for** $t = 1 \dots T_{plan}$ **do**
    Perform subgoal update $\text{ix}_t = f(\text{ix}_{t-1}, I_{t-1}, K_{1:L}^*)$.
    Initialize latents from prior: $\boldsymbol{u}_0 \sim \mathcal{N}(0, I)$.
    **for** $i = 0 \dots N_{it}$ **do**
        Rollout prediction model for $l$ steps, obtain $M$ future sequences $\hat{\boldsymbol{I}}_{t:t+l}$.
        Compute distance between final and subgoal image: $c = \text{dist}(\hat{I}_{t+l}, K_{\text{ix}_t})$.
        Choose $M'$ best sequences, refit Gaussian distribution: $\boldsymbol{\mu}_{i+1}, \boldsymbol{\sigma}_{i+1} = \text{fit}(\boldsymbol{u}_i')$.
        Sample new latents from updated distribution: $\boldsymbol{u}_{i+1} \sim \mathcal{N}(\boldsymbol{\mu}_{i+1}, \boldsymbol{\sigma}_{i+1})$.
    **end for**
    Execute $u_{N_{it},0}^*$ and observe next image $I_t$.
**end for**

---

The parameters used for our visual planning experiments are listed in Supplemental Table 8.

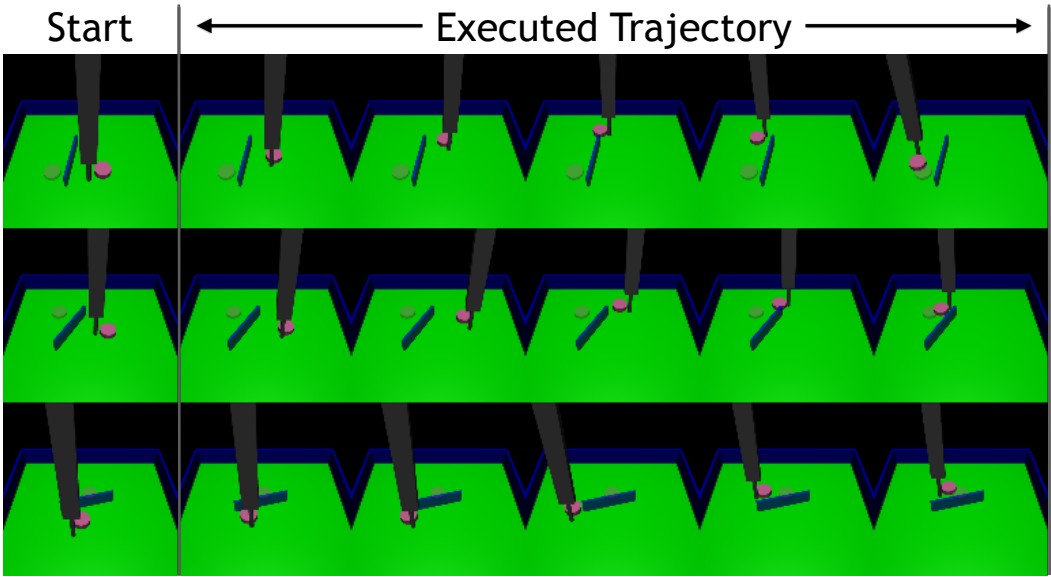

Figure 13: Sample planning task executions from the test set. From a start state depicted on the left, the robot arm successfully pushes the object into the goal position (semi-transparent object) guided by the KEYIN subgoals. The right side of the figure shows intermediate frames of the execution trajectories.

