# OpenReview forum: "Keyframing the Future: Discovering Temporal Hierarchy with Keyframe-Inpainter Prediction"
_ICLR.cc/2020/Conference — Reject_

### Official Review · AnonReviewer1 · 2019-10-23
**Official Blind Review #1**

**Rating:** 3

**Review:**

The paper introduces a model trained for video prediction hierarchically: a series of significant frames called “keyframes” in the paper are first predicted and then intermediate frames between keyframes couples are generated. The training criterion is maximum likelihood with a variational approximation. Experiments are performed on 3 different video datasets and the evaluation is performed for 3 tasks: keyframe detection, frame prediction and planning in robot videos.
The idea of generating an abstraction or a summary of a video via a sequence of important frames is attractive and could probably be used in different contexts. The proposed model is new and the authors introduce some clever ideas in order to train it. The evaluation work is important and the authors propose different settings for this evaluation.
The paper also present weaknesses. First the motivation for keyframes generation should be better developed: the model does not perform better than baselines for video frames prediction so that keyframes generation should be motivated by other applications. Planning as proposed by the authors could be one, but in this case it should be more developed. The main weakness is however the technical presentation which is painful to follow. When it is possible to get a general picture of what is done, it is quite difficult to figure out exactly how the model works. A global rewriting and maybe a better focus are required for publication. The probabilistic model (section 3.1) is relatively clear, even if it could be improved. It seems that the generation of a keyframe and the prediction of the corresponding time (tau^n)  are independent (eq. 3). This could be commented. Also it seems that in eq. 3 the log(K|z..) term should be inside an expectation. Section 4 was difficult to decipher for me. My understanding is that instead of sampling from a multinomial during training, you bypass this non differentiable operation by using what you call “soft targets” thus obtaining a differentiable objective (eq. 4). Is that true? In any case, the procedure should be made a lot clearer. The “intermediate frame” passage also remained confuse for me.
Considering the experiments, the authors make an important effort in order to evaluate different aspects of their model. In a fisrt step, they evaluate the ability of the model to generate significant keyframes using a detection setting.  It is not clear how they define ground truth frames for this evaluation. Those ground truth frames are defined as the frames where the movement in the image changes, which is easy on the Brownian movement dataset but what about the others? Also the baselines used in this comparison are weak. In the paper of Denton, they suggest some way to detect surprise and apparently this is not what you used. This should be justified/ commented. For keyframe modeling the proposed model behaves similarly to the baselines and even performs worse than the simpler “jumpy” model. Concerni g the paragraph about the selection of the number of predicted keyframes, it is not clear what is the reference (ground truth) number of target keyframes.
 The planning experiments are interesting, but difficult to follow at least from the main text.
Overall, I think that there are several interesting ideas and realizations. They should be better put in perspective and explained.

----- post rebuttal -----------

Thanks for the detailed answer. The paper is largely improved both for the style and the comparisons. But still requires further improvements. I will keep my score.


**Experience Assessment:**

I have published one or two papers in this area.

**Review Assessment: Checking Correctness Of Derivations And Theory:**

I assessed the sensibility of the derivations and theory.

**Review Assessment: Checking Correctness Of Experiments:**

I carefully checked the experiments.

**Review Assessment: Thoroughness In Paper Reading:**

I read the paper thoroughly.

---

> ### Author Response · Authors · 2019-11-13
> **Part 2 - additional improvements / clarifications**
>
> Below we respond to additional reviewer’s comments regarding clarity. We hope that this addresses the reviewer’s points and would be happy to incorporate any further suggestions for clarification.
>
> == Not clear how ground truth keyframes are defined ==
>  As the reviewer points out, we define keyframes as the points of motion change in the Stochastic Brownian Motion dataset. For the robot pushing dataset, keyframes are those frames when the robot lifts its arm to transition between pushes. For the gridworld dataset, frames in which the agent interacts with an object are defined as keyframes. This is explained in the third paragraph of Sec. 6.1 in the submission. We refer the reviewer to the supplementary website with visual examples of the trajectories. To further clarify this, we expanded the description in the manuscript.
>
> == Keyframe and \tau distributions independent ==
>  The keyframe and the \tau distributions are indeed modeled as independent, given the latent variable. This is desirable as we want the latent variable to describe the dependencies between these variables. We added a footnote to Eq. 3 for clarification.
>
> == Missing expectation around keyframe likelihood ==
>  We thank the reviewer for pointing out this typo, the log(K|z,I_co) term in Eq. 3 should indeed be inside the expectation over q(z), we corrected this.
>
> == Planning experiments hard to follow ==
>  We thank the reviewer for pointing out the need for further clarification. In the submission we described the details of the planning procedure in Sec. K of the appendix and in Alg. 2,3. To help clarify the planning experiments in the main text we included several sentences in Sec. 6.4 detailing the employed planning procedures.
>
> We hope that the changes to the manuscript address all points raised in the review. Please let us know if there are any other points that should be addressed and otherwise please consider updating your review.
>
> [1] Self-supervised visual planning with temporal skip connections, Ebert et al., CoRL 2017
> [2] Visual foresight, Ebert et al., 2018
> [3] Time-Agnostic Prediction, Jayaraman et al., ICLR 2019
> [4] Dynamics-Learning with Cascaded Variational Inference for Multi-Step Manipulation, Fang et al., CoRL 2019

---

> ### Author Response · Authors · 2019-11-13
> **Performed experiment with surprise baseline, keyframe modeling, improved presentation, clarified motivation**
>
> We thank the reviewer for the helpful comments and suggestions. We have made several changes to the presentation of the technical section as well as the description of the experiments and results to address the reviewer's concerns (updates in red). We have also added comparisons to alternative “surprise” baselines as well as experiments on keyframe modeling. We hope the responses below address the reviewer’s concerns and we are happy to make additional changes if those are required.
>
> == Motivation should be better developed ==
> We thank the reviewer for pointing out a possible confusion about the motivation of the paper. We tried to highlight in the original submission that the goal of this work is not to improve video prediction quality but instead to discover meaningful temporal structure in sequences. As the reviewer notes, one possible application for the temporal structure discovered by our model is predicting subgoals for efficient long-horizon planning, and we now expanded the introduction to discuss this. We note that planning is known to be a challenging task [1-4] and we show that our model is able to outperform strong baselines using the learned temporal abstraction.
>
> == Keyframe detection baseline weak, use Denton&Fergus’18 ==
>  We thank the reviewer for proposing this alternative comparison. In the original submission we used a measure of surprise based on the KL divergence (see Sec. 6.3, details in appendix, Sec. F). To support the strength of this baseline we have now added two baseline evaluations using alternative definitions of surprise. The first uses the method of Denton&Fergus’18 that the reviewer proposed. The second uses the lower bound on the log-likelihood -log(p) instead of just the KL divergence to measure surprise. In Tab 3,4, we find that the KL-based baseline reported in the submission consistently performs on par with these alternative formulations, and that our method outperforms all baselines.
>
> == No performance gain on keyframe modeling ==
> We understand that the reviewer is referring to the results evaluating image sequence prediction quality (as opposed to keyframe prediction quality). It is true that our method does not improve performance on video modeling, but we emphasize that it is able to perform on par with recent work (Denton&Fergus’18) while additionally discovering the keyframe structure of the sequence. Please refer to our answer to the first question about motivation. We additionally performed an experiment showing that our model’s ability to model keyframes (i.e. the most important frames) is superior to the baseline methods that do not discover temporal structure in Tab. 6. We further clarified the emphasis of the paper with an additional sentence in Sec 6.2.
>
> == Soft targets for obtaining differentiable objective? ==
>  This interpretation is correct: we introduce the relaxed formulation to bypass the sampling step from p(tau|z,I_co) and make the formulation fully differentiable. The original submission stated this in Sec. 4. We thank the reviewer for pointing out the need for further clarifications. We restructured Sec. 4 to more clearly motivate and derive the soft relaxation objective. We additionally improved Fig. 3 to better illustrate the procedure.

---

### Official Review · AnonReviewer2 · 2019-10-26
**Official Blind Review #2**

**Rating:** 3

**Review:**

This paper introduces a variational objective to train a model which can jointly select keyframes of a video and generate the intervening frames to produce a resultant video. The model is provided an initial set of frames as context. At training time the model always learns to produce N*J frames, where N is the number of keyframes and J is a fixed number of frames to generate for each keyframe. The authors compare their method for selecting informative keyframes on a number of baselines and show an improvement over these baselines. The problem is interesting and well-motivated, but I have some concerns with the proposed approach and experiments. As such, I am a weak reject.

comments / questions:
- Equation 3 lacks context. Initially, when looking at the authors' objective it seems that the inner expectation should be taken with respect to the joint time indices for the current and next keyframe. Only later after equation 4 do they mention that they always predict a fixed number of frames J.
- The need for normalizing over the first T timesteps in equation 4 seems quite messy. Is it guaranteed that all of the needed keyframes will actually be within the first T timesteps? How does this work in practice?
- Many important details of the inference procedure are relegated to the appendix. For example, there are no details for extracting which of the 60 keyframes that were trained for a sequence (due to the fixed length sequences) should be selected at test time. Looking at the appendix, it is clear that the approach requires an extensive planning algorithm at inference time, which seems like an important component.
- The authors prominently highlight that their method is fully differentiable, yet they train in two stages while freezing weights. Why isn't the model trained end-to-end? The stated reason for doing so is that this "simple" two-stage procedure improves optimization. What exactly happens if you don't do this two stage training process? Does it fail to learn? Some experimental numbers would be nice to see.
- The authors do not compare their method to any strong keyframe prediction baselines. Considering there is existing work in keyframe prediction, it seems important to highlight the difference between other competing models, rather than relying on simple baselines. Why don't they use self-information/surprisal as a baseline i.e., by training an autoregressive model on the frames and then picking the N frames with the largest -log(p)? This is a metric that has been investigated frequently and has better interpretability than defining a new measure of surprise. Note that Kipf et al. (2019) uses this notion of surprisal as well.
- Sauer et al. (BMVC 2019) should likely be cited as it does very similar keyframe analysis. Also, as the ICML 2019 conference had already concluded by the ICLR submission deadline, is it really fair to state the work with Kipf et al. (2019) was conducted in parallel?
- Why does the model trained to learn a fixed number of timesteps for the intermediate frames? Did they investigate jointly predicting the indices for the current and next timesteps? It seems like it would greatly simplify their inference scheme if they did this. If they tried that approach and it failed, maybe that should be mentioned in the paper (with an explanation as to why it fails).
- In the literature review, when discussing hierarchical temporal structure, the authors state: "However, these models rely on autoregressive techniques for text generation and are not applicable to structured data, such as videos." Autoregressive techniques have been investigated in relation to videos; in fact, the authors later describe papers that have used autoregressive techniques for modeling videos.


**Experience Assessment:**

I do not know much about this area.

**Review Assessment: Checking Correctness Of Derivations And Theory:**

I assessed the sensibility of the derivations and theory.

**Review Assessment: Checking Correctness Of Experiments:**

I assessed the sensibility of the experiments.

**Review Assessment: Thoroughness In Paper Reading:**

I read the paper at least twice and used my best judgement in assessing the paper.

---

> ### Author Response · Authors · 2019-11-13
> **Part 2 - additional improvements / clarifications**
>
> == 1. Equation 3 typo? ==
> The expectation in Eq 3 is indeed taken with respect to the current and next keyframe indices. We apologize for the typo where only the current keyframe index was specified: we have corrected this in the manuscript.
>
> == 2. Why normalize over T frames/all keyframes in first T? ==
> We only use the first T frames output by the network as the prediction, and we do not enforce that all of the keyframes predicted by the network fall in the first T frames. Keyframes predicted after the first T total frames are discarded. Our network thus can avoid using its full keyframe budget by putting keyframes after the predicted sequence. In practice, we observe that this feature allows the network to discard extraneous keyframes when a sequence is well modeled with a fewer number of keyframes, such as in the bouncing ball experiments in Fig 9.
>
> == 4. Why two-stage training when the model is fully differentiable? ==
> We thank the reviewer for this insightful question. In our initial experiments we observed that without the two-stage training the network failed to predict sequences that are long enough. As described in Sec 5.2, we hypothesize that pre-training the inpainting network aids optimization as it learns inpainting strategies for a variety of different inputs, allowing it to generate longer sequences. We note that end-to-end differentiability is important as we utilize it in the second stage of the training to backpropagate the error of the keyframe predictor through the inpainter.
>
> == 6. Cite Sauer; Kipf concurrent? ==
> Sauer’19 (Tracking Holistic Object Representations) does not appear to perform keyframe analysis. The work of Kipf’19 was conducted in parallel to us as evidenced by preprint version of our paper (which we do not link for the purposes of double-blind review) that was cited by Kipf’19 as concurrent work.
>
> == 7. Try to jointly predict current, next timestep? ==
> The indices for the timesteps of the current and the next keyframe are indeed predicted by the same network as described in Sec 5.1.
>
> == 8. Autoregressive video models. ==
> We thank the reviewer for pointing this out and we have modified the original statement. While it is possible to adapt models like HMRNN to videos by using autoregressive prediction, as stated in our related work summary, autoregressive models for video prediction suffer from slow inference, and take minutes to generate a video even for the fastest models. We thus believe this approach to keyframe modeling would be impractical.
>
> We hope that we were able to address all points raised by the reviewer. Please let us know if there are any other points that should be addressed and otherwise please consider updating your review.

---

> ### Author Response · Authors · 2019-11-13
> **Provided clarifications, updated presentation, performed experiment with surprisal baseline**
>
> We thank the reviewer for the useful comments and suggestions. We understand that the main concerns of the reviewer are that 1) we only predict J intermediate frames, 3) the inference procedure is unclear, and 5) the surprisal baseline is not the same as in Kipf’19. There appear to be several critical points of misunderstanding, which we will try to clarify. While we made our best attempt to address the reviewer’s concerns, we hope the reviewer can engage with us during the discussion period to clarify their concerns and help us understand how we can address them. We updated and clarified parts of the technical section based on the reviewer’s feedback (updates in red).
>
> We next address the reviewer’s concerns, numbered by the order in which they appear in the review, starting with the most crucial.
>
> ==  1, 7. Predict J intermediate frames? ==
> The number of frames between two keyframes is not fixed - it is given by the variable tau as explained in Sec. 3.1, eqs (1, 2). However, in practice, the number of frames between the keyframes is indeed bounded by J (for computational reasons). We note that at training time, we always generate all J frames as the timestep distribution tau in general has non-zero mass at each timestep. However, at test time we only predict argmax(tau) intermediate frames.
>
> == 3. Missing details of inference? Planning needed for inference? ==
> We interpret the question as asking about the test time procedure of the model (rather than the variational inference procedure we use). At test time, we can use our model in three different ways, to (1) generate a sequence given conditioning frames, as described in Sec 3.1, (2) infer the keyframes of a sequence, by using the inference network described in Sec 3.2, or (3) perform planning to reach a goal as described in Sec 6.4. We suspect the reviewer’s question is about (1), which is performed as follows. We sample the latents from the prior distribution p(z|I_co), generate the corresponding keyframes K and their indices tau, and use the inpainter network to generate the rest of the sequence. We note that the planning algorithm described in the appendix is used for motion planning as part of (3) and is not a part of the training. Planning is not required for (1) or (2).
>
> == 5. Why this surprisal metric? ==
> As the reviewer points out, our original surprisal metric is not the same as the traditional log(p). While a lower bound on log(p) can be computed by summing the reconstruction and the KL-divergence loss, we only used the KL-divergence part. We have now added a version of the baseline that computes the full lower bound as well as the version from Denton&Fergus’18 suggested by R1. In Tabs. 3, 4, we find that these new versions of the surprise baseline perform comparably to the one we originally reported, and our method outperforms all of them.

---

### Official Review · AnonReviewer3 · 2019-10-28
**Official Blind Review #3**

**Rating:** 6

**Review:**

The authors address the problem of discovering and predicting with hierarchical structure in data sequences of relevance to planning. Starting with the kinds of data that have been used recently in video prediction, the authors aim at learning a sequence of keyframes (i.e., subsets of frames forming the overall sequence) that in a suitable sense "summarize" the overall trace. As they rightly note, many alternate models struggle with making good long term predictions in part because they focus on all levels of prediction equally.

The technical approach is to pose the problem as one of inferring the temporal location of each of these key frames and then to interpolate with a model to generate intermediate frames. One could try to make either step sophisticated - the authors choose to make the keyframe selection more sophisticated and interpolation simpler. The paper first described the KeyIn model in terms of a probabilistic model of jointly finding the Ks and then the inpainted Is. This can become delicate, so the authors propose a relaxation that is more forgiving when the keyframe locations are being searched for. Learning is driven by a reconstruction loss of finding the approximate location, locally interpolating and then seeing if this accords with the training data. This is all implemented with an LSTM based NN architecture which seems sound to me.

I feel the paper is taking on the right kinds of questions, looking for ways to inject the right kind of structure. I do have some concerns about the overall formulation:

1. Much of the paper is focussed on rather clean images where nothing extraneous is happening. In reality, the backgrounds of real images is not so benign and other extraneous dynamics might interfere. While I understand this is a step towards the long term goal, I wonder if the end result is a bit too incremental in the absence of some attempt to explore this source of (lack of) robustness.

2. In §6.3, the authors try to demonstrate that the number of keyframes parameter can be wrong by a little bit but these are still small ranges. In realistic images it is likely that the total number of keyframes selected by such an algorithm is much larger due to extraneous events. This is why a proper robustness study is crucial on more realistic input. As it, in anything other than the trivial dot on black background, the precision-recall numbers are fairly modest. This will likely degenerate into noise in most camera-based images of the kind seen by a real robot. So, how much confidence should we expect to have in the approach's generality?

3. For the baselines, the true good baseline might have been a human annotation that tells us how people really conceptualise the structure. With data such as pushing, this might not be so different from the simple visual inspection, but again with real data this will vary. The paper would really be much stronger if these were addressed.




**Experience Assessment:**

I have published one or two papers in this area.

**Review Assessment: Checking Correctness Of Derivations And Theory:**

I assessed the sensibility of the derivations and theory.

**Review Assessment: Checking Correctness Of Experiments:**

I assessed the sensibility of the experiments.

**Review Assessment: Thoroughness In Paper Reading:**

I read the paper thoroughly.

---

> ### Author Response · Authors · 2019-11-13
> **Performed an experiment with gaussian noise**
>
> We thank the reviewer for the helpful comments and suggestions. We made the following changes to the submission to address the reviewer's remarks and answer the posed questions.
>
> == robustness in noisy settings ==
> We thank the reviewer for drawing attention to the importance of robustness to noise of the kind that exists in real-world domains. We added an experiment (Tab. 7 and Fig 11) showing that KeyIn keyframe detection performance is robust to Gaussian image noise, a noise characteristic that is commonly found in real camera sensors [5]. We believe this experiment provides some initial evidence that KeyIn can learn representations that are robust to noise. We note that comparable prior work uses environments with little noise and few distractors [1,2,3,4], but we hope that future work will be able to investigate this direction further once the video modeling community matures to the point of using complex real-world data with diverse background activity of the kind the reviewer suggests.
>
> == Modest precision-recall numbers ==
> The reported precision-recall numbers indeed look modest, however, this is largely due to the inherent ambiguity when defining “true” keyframes. For example: in the grid-world environment, is the frame where the agent reaches an object or where it interacts with the object the better keyframe? While we chose one definition of a keyframe, we find that the method often predicts keyframes consistent with another definition, leading to off-by-one errors that are severely penalized by precision-recall metrics. To address this, we added an evaluation that measures minimum temporal distance to the true keyframe in Tab 4, which shows that KeyIn places keyframes closer to the annotated keyframes than all baselines. We also point out that the planning experiments provide a more objective evaluation metric for the quality of keyframes, and we find that KeyIn improves planning performance over all baselines.
>
> == Human annotation baseline ==
> We thank the reviewer for this suggestion. We agree that on the current environments the human annotations would not be much different from the ones used to evaluate the models. However, for future work that extends KeyIn to more complex environments where it is even harder to define “objective” keyframes, crowdsourced human annotations will likely be necessary for proper evaluation.
>
> [1] Time-Agnostic Prediction, Jayaraman et al., ICLR 2019
> [2] Stochastic Video Generation with a Learned Prior, Denton&Fergus, ICML 2018
> [3] Learning latent dynamics for planning from pixels, Hafner et al., ICML 2019
> [4] Robustness via Retrying: Closed-Loop Robotic Manipulation with Self-Supervised Learning, Ebert et al., 2018
> [5] https://en.wikipedia.org/wiki/Image_noise

---

### Author Response · Authors · 2019-11-13
**Updated presentation of the approach, performed experiments with noise, surprise, keyframe modeling.**

We thank the reviewers for their feedback. We note that the reviewers found the addressed problem of keyframe discovery interesting and well-motivated and the proposed method novel. At the same time, the reviewers requested clarifications, suggested further analysis and improvements to the presentation. We performed additional experiments and otherwise updated the manuscript according to the suggestions (updates in red). We summarize the comments we believe are most crucial and respond to them here.

R3: An evaluation on a dataset with noise from background activity would strengthen the paper.
A: We added an experiment showing robustness of our method to background noise activity on Pushing and Gridworld data in Tab 7. We believe this quantitative evaluation elucidates some properties of the method, and we hope that future work will be able to investigate this further once the video modeling community matures to the point of using complex real-world data with diverse background activity of the kind the reviewer suggests.

R1,2: Some prior work uses a different definition of surprisal than the one we use as a baseline.
A: Our original submission contained a version of a surprisal baseline measuring an approximation to log(p(s)). We have now added a version of this baseline of the same form as the one in Kipf'19 suggested by R2 as well as the version from Denton&Fergus’18 suggested by R1. We find that these new versions of the surprise baseline perform comparably to the one we originally reported, and our method outperforms all of them in Tab 3,4.

R1: Including more details about the planning experiment in the main paper would make it easier to follow.
A:  We added further details to the planning section in the main body of the paper. We note that this section spans almost a page and for the reasons of space we left certain details of the procedure in the supplement. We will happily move specific parts of the supplement to the main paper if they are deemed necessary to understand this section.

R1: The proposed method does not beat prior methods when evaluated on video prediction, other motivation is needed.
A: Indeed, we motivate the proposed method for keyframing the future as 1) being able to build a compact representation of the sequence just in terms of discovered keyframes on three datasets, and 2) being further able to utilize such representation for planning, empirically surpassing strong prior work. Additionally, we added an experiment showing that our model outperforms baselines on *keyframe* modeling, showing that it is better than prior methods on predicting the important frames in the sequence (while being on par for sequence modeling) in Tab 6.

---

### Author Response · Authors · 2019-11-15
**All comments addressed and waiting for feedback**

We have yet to hear back from the reviewers, and our window to reply is rapidly closing. We appreciate any additional comments you have. Thank you!

---

### Decision · Program_Chairs · 2019-12-19

**Decision:**

Reject

**Comment:**

The paper is interesting in video prediction, introducing a hierarchical approach: keyframes are first predicted, then intermediate frames are generated. While it is acknowledge the authors do a step in the right direction, several issues remain: (i) the presentation of the paper could be improved (ii) experiments are not convincing enough (baselines, images not realistic enough, marginal improvements) to validate the viability of the proposed approach over existing ones.